# Oral pre-exposure prophylaxis retention among men who have sex with men and transgender persons: Systematic review and meta-analysis

Feline de la Court[1], David O. T. ten Hoff[1†], Maria Prins[1,2,3], Elske Hoornenborg[1], Maarten F. Schim van der Loeff[1,2,3], Liza Coyer[1,2,3☺], Anders Boyd[1,2,3,4☺]*

1 Department of Infectious Diseases, Public Health Service of Amsterdam, Amsterdam, the Netherlands, 2 Department of Infectious Diseases, Amsterdam institute for Infection & Immunity (AII), Amsterdam UMC, University of Amsterdam, Amsterdam, the Netherlands, 3 Department of Infectious Diseases, Amsterdam UMC, University of Amsterdam, Amsterdam, the Netherlands, 4 Stichting HIV Monitoring, Amsterdam, the Netherlands

† Deceased
☺ These authors contributed equally to this work.
* a.c.boyd@amsterdammumc.nl

## Abstract

Successful implementation of oral pre-exposure prophylaxis (PrEP) requires retention in PrEP care among those with an increased likelihood of HIV. This study aimed to estimate the proportion of retention on PrEP and extent to which variability in PrEP retention is associated with population-, program-, and study-specific characteristics among men who have sex with men (MSM) and transgender persons (TGP). We performed a systematic review and meta-analysis examining PrEP studies and conference abstracts retrieved from the PubMed and Ovid online databases, capturing demonstration projects or observational studies published from January 1, 2010 to March 24, 2021. We included 84 studies (totaling 90 study "cohorts" analyzed) that reported on the retention of oral PrEP for HIV and included predominantly MSM and TGP. The proportion of retained participants was obtained from each study and used to estimate the cumulative probability of being retained on PrEP and the loss of PrEP retention rate over time via a random-effects meta-analysis survival model. We examined sources of heterogeneity by including study-level covariates in this model. The pooled cumulative probability of PrEP retention was 77.0%, 64.7%, 48.5%, and 24.1% at 6, 12, 24, and 60 months, respectively. Loss of PrEP retention rates were significantly (p < 0.05) lower in studies from Europe, Australia and multiple regions (vs. North America), and with ≥3-monthly follow-up (vs. < 3). Rates were significantly higher in studies from Africa (vs. North America), with lower median age, higher proportions of non-MSM/TGP participants, higher proportions of participants with unspecified (vs. mixed) ethnicity, with unspecified (vs. daily) PrEP regimen, free-of-charge (vs. unspecified) STI testing, whose data were from conference abstracts (vs. peer-reviewed papers), and that were more recent. In conclusion, oral PrEP retention

**Data availability statement:** The minimal dataset underlying the findings of this study is publicly available in a GitHub repository at: https://github.com/andersboyd2/GGD_PrEP_retention. All relevant data and code used in analysis are included. The repository will remain publicly accessible and archived for reproducibility purposes. All authors have agreed on this data sharing plan.

**Funding:** The current study is part of the OptiPrEP (optimizing pre-exposure prophylaxis roll-out among men having sex with men) project, funded by the Aidsfonds (P-54601). The funders had no role in study design, data collection and analysis, decision to publish, or preparation of the manuscript.

**Competing interests:** MP obtained unrestricted research grants and speaker/ advisory fees from Gilead Sciences, Abbvie and MSD; all of which were paid to her institute and were unrelated to the current work. EH obtained unrestricted research grants from Gilead Sciences, which were paid to her institute and were unrelated to the current work. The institution of MFSvdL receives study funding from GSK; he served on advisory boards of GSK and Merck/MSD; fees were paid to his institution. AB has received speaker fees from Gilead Sciences, which was unrelated to the current work. This does not alter our adherence to PLOS ONE policies on sharing data and materials. The other authors report no conflicts of interest.

decreased over time and differed across population-, program-, and study-specific characteristics. The heterogeneity across studies highlights PrEP implementation challenges and the need for tailored retention strategies.

## Introduction

Oral pre-exposure prophylaxis (PrEP) is a highly effective prevention strategy against HIV. PrEP can be taken either every 24 hours (i.e., daily) or before and after sexual contact (i.e., event-driven). Its successful implementation requires that individuals are retained in PrEP care, particularly those with higher likelihood of acquiring HIV [1]. Retention can refer to many things, but is generally considered as sustained engagement in PrEP care when one's likelihood of HIV infection is increased [2]. Adherence to oral PrEP, specifically at levels allowing effective prevention of HIV, could also reflect retention. Men who have sex with men (MSM) and transgender persons (TGP) are important target populations for PrEP considering their increased risk HIV acquisition [3]. However, preliminary epidemiological evidence suggests that the proportion of individuals from these groups who are retained in PrEP care is suboptimal, especially in the long term [4].

Given that the incidence of HIV is particularly high among those who discontinued or had gaps in their oral PrEP use [5,6], low retention in PrEP care is concerning [5]. A previous systematic review on PrEP retention among MSM and TGP has shown a pooled proportion of PrEP discontinuation of 32% at 6 months and 35% up to 12 months [7]. That review evaluated 59 studies, but data around and beyond 12 months of follow-up were scarce. Furthermore, research has suggested that retention is often lower among MSM and TGP who are adolescent or in early-adulthood, socioeconomically disadvantaged or from ethnic minority groups [5,8]. Retention outcomes may also be attributable to factors related to location, financing and implementation or research in PrEP care [9–11]. Retention outcomes could also vary based on different definitions of retention (e.g., PrEP continuation, follow-up in care), and these definitions will have to reflect the type of oral PrEP being used (i.e., daily or event/driven). For example, if retention is defined by follow-up in care (i.e., still receiving STI testing, PrEP refills or new prescriptions), this criterion could apply to both oral daily and event-driven PrEP. However, for an individual who uses event-driven PrEP and may discontinue PrEP because they are no longer engaging in activities associated with HIV acquisition, a definition based on PrEP continuation would need to consider this risk and eventually whether the person is still engaged in care. So far, no previous review study has provided an aggregated analysis of the extent to which such program-specific characteristics, including specific definitions of retention, affect PrEP retention outcomes.

This systematic review and meta-analysis estimated the proportion of retention on oral PrEP of studies that predominately included MSM and TGP. We also analyzed the extent to which variability in the loss of PrEP retention is associated with population-, program-, and study-specific characteristics. Such insights may contribute to strategies for effective and sustained oral PrEP care.

## Methods

### Study design

We performed this systematic review and meta-analysis in accordance with guidelines from the Cochrane Handbook for Systematic Reviews of Interventions [12] and described this analysis using the Preferred Reporting Items for Systematic Review and Meta-analysis guidelines [13] (S1 Appendix). This study was not registered in a register for systematic reviews.

### Search strategy and inclusion criteria

A systematic search for peer-reviewed papers and conference abstracts was conducted using the PubMed and Ovid online databases, retrieving studies published from January 1, 2010 to March 24, 2021. The full search strategy is given in S1 Table. Additional abstracts were retrieved from conference databases of the International AIDS Conference and the Conference on Retroviruses and Opportunistic Infections from 2017–2021. We used EndNote X9.3.3 to manage all citations. An initial selection was made based on title and abstract, and two independent reviewers (F.d.l.C. and L.C.) conducted a full-text screening for study inclusion.

We included studies conducted in predominantly MSM and/or TGP populations (i.e., ≥60% of the study population consisted of one of these key populations) who were using oral PrEP for HIV (i.e., fixed dose combination of emtricitabine and tenofovir disoproxil fumarate) taken daily, event-driven or both. We did not include studies exclusively without MSM or TGP because the barriers and facilitators to PrEP retention are likely different for these other key populations [14] and could increase heterogeneity. No comparison between regimens was required for inclusion. Studies were required to provide a quantitative measure of retention on PrEP. We categorized definitions of retention related to the following: PrEP continuation, where retention was defined by whether someone was still on PrEP according to self-reports, prescription data, etc.; follow-up, where retention was defined by whether participants were still actively in care or enrolled in a study; mixed/other. Studies were included if follow-up time and proportion of participants retained from PrEP initiation to end of follow-up were reported or could be calculated for at least one of the pre-defined follow-up time points (i.e., beginning at month 3 since PrEP initiation, and continuing for every 3 months until month 60). Eligible study designs were unblinded randomized controlled trials (RCT), open label extension (OLE) studies, longitudinal observational cohort studies, demonstration projects, implementation projects, and studies that analyzed routine care data from primary care or other public health settings. Double- and single-blinded RCTs were excluded to more appropriately approximate real-world settings [15]. Studies were also excluded if follow-up parameters were unclear (i.e., follow-up did not begin at start of PrEP use or only median follow-up time was reported) or not mentioned. No restrictions were imposed based on language or geographical location. If multiple studies described the same study population or cohort, the study with the longest follow-up time and most complete follow-up information was included. For open-label RCTs containing both an "intervention" and "standard of care" arm, each arm was included as a separate study cohort if the arm included individuals using PrEP. If the arm did not include individuals using PrEP, it was not included in the study.

### Outcome

The primary outcome was retention on PrEP, as defined above, at each of the pre-defined time points (every 3 months from month 3 to month 60). This proportion was calculated by dividing the number of participants reported to be retained on PrEP by the total number of participants who initiated PrEP in the study.

### Data extraction

Data from all included studies were independently extracted by two reviewers (F.d.l.C. and L.C.) using a standardized form. Discrepancies in data were discussed between reviewers and the most appropriate data to include were determined through consensus. Consensus was achieved for all discrepancies.

Data were extract on the following: region (i.e., North America, Europe, Australia, South America, Africa, Asia, or multiple); country income from the World Bank (i.e., regions were classified as low to middle income, high income, or multiple income); calendar years at the beginning, completion, and publication of study; study population (i.e., MSM [when >80% is MSM] or "MSM, TGP and other"); percentage of MSM, TGP, or other in the study population; geographic setting (i.e., urban, rural, or mixed); ethnicity of study population (i.e., mixed, Black only, Asian only, unspecified); median age; frequency of follow-up visits (i.e., every 3 months or more frequently, or less frequently than every 3 months); outcome definition (i.e., continuation, follow-up, or mixed/other); publication type (i.e., paper or conference abstract); PrEP regimen (i.e., daily, event-driven, both, or unspecified); PrEP and STI testing costs for the participant (i.e., completely free-of-charge, not completely free-of-charge, or unspecified). We included the "unspecified" category to assess missing or unclear results in the synthesis of the latter two variables.

## Risk of bias assessment

The methodological quality and risk of bias of included studies was assessed by one reviewer (F.d.l.C.) using the Newcastle-Ottawa Quality Assessment Scale for cohort studies. Since we analyzed RCTs as cohort studies, these were also evaluated using the Newcastle-Ottawa Quality Assessment Scale with some adaptations (S2 Table).

## Statistical analysis

We used the proportion of participants reporting to be retained on PrEP to model the cumulative probability of PrEP retention over time. This probability could be estimated for an individual study $i$, defined as the survival probability, $S_i$, using a random-effects model similar to that described in Arends et al. [16]. Reported survival probabilities were transformed as $\ln(-\ln(S_i))$ and were regressed on a fixed-effect parameter including time and a vector of random coefficients and its corresponding design matrix. This design matrix contained an intercept and time effect, whose variance-covariance structure is unstructured.

The error term was modeled as a column vector of residuals containing a covariance matrix, V, whose main diagonal was set equal to the squared standard errors of the survival probabilities. These standard errors were calculated using Greenwood's formula and were made to correspond to the $\ln(-\ln)$ survival probabilities using the formula in Arends et al. [16]. Survival probabilities from the same study can be considered correlated, whose covariance is iteratively estimated as blocks of within-study covariance matrices within the covariance matrix, V. However, the model failed to achieve convergence and we were thus unable to account for within-study correlation.

The model was used to generate the predicted cumulative probability of PrEP retention and its 95% confidence interval (CI). We estimated the model using the "meglm" command in STATA. In sensitivity analyses, we generated separate curves while only including randomized control studies (to assess bias from observational data), studies with more than 100 participants (to assess the effect of potential publication bias), and studies with a Newcastle-Ottawa Quality Assessment Scale risk score bias of more than 5. Given that most studies collected data until month 24 of follow-up, we also generated the curve using data up until month 24 to assess any change in model fit. The model was slightly modified for some of these sensitivity analyses for purposes of convergence: variance-covariance structure was exchangeable when including only randomized-control studies.

To assess the sources of heterogeneity, we could estimate the proportion of PrEP retention over time using the same random-effects survival model and compare proportions across levels of study characteristics by including additional covariates in the fixed-effect parameter of the model. The resulting parameter estimate is a hazards ratio (HR) and 95%CI comparing the rates of loss of PrEP retention across levels of study variables. This analysis is referred herein to as meta-regression. To determine whether HRs changed over calendar years, we included calendar year of study start and the interaction term between calendar year of study start and factor to each univariable model, separately, and tested the interaction term.

As several diagnostic statistics have not yet been developed for the random-effects survival model approach, we employed a meta-analysis random-effects model with between-study variance calculated using restricted maximum likelihood estimation [17]. This model was run for each of the timepoints at which the majority of studies reported data (i.e., months 3, 6, 9, and 12). We used the "rma" function of the "metafor" package in R to estimate the model. From this model, we calculated the measure of heterogeneity $I^2$ and tested the null hypothesis of no between-study heterogeneity using the Q statistic at each timepoint. We evaluated whether certain combinations of studies were contributing to excessive heterogeneity using the graphical display of study heterogeneity (GOSH) plot [18]. A total of 10,000 combinations were randomly selected to produce the graph using the gosh() function in the "metafor" package. We assessed the influence of outlying studies at each timepoint using the method developed by Viechtbauer and Cheung [19] in the "metafor" package. We also assessed for publication bias at these same timepoints using the Egger test of bias [20] with $p < 0.05$ indicating significant publication bias.

We carried out analysis using STATA (v15.1, College Station, TX, USA) and R (v4.0.3, Vienna, Austria). We defined significance as $p < 0.05$.

## Results

### Characteristics of included studies

Fig 1 presents the flow-diagram for study selection. The database search identified 6853 studies published between 2010 and 2021. After removal of duplicates (n = 3627) and exclusion based on title and abstract (n = 2760), 466 full-text studies were screened. From these, 382 studies were excluded for reasons which are summarized in Fig 1. Finally, 84 studies were selected (S1 References). Of these, 6 were open-label RCTs containing both an "intervention" and "standard of care" arm; each of these arms were considered as separate study cohorts and hence 90 study cohorts were included in the analysis. The "standard of care" arms for these studies included PrEP.

The 90 study cohorts were reported in 53 peer-reviewed papers and 31 conference abstracts. The largest proportion were cohort studies (n = 41; 45.6%), followed by demonstration or implementation projects (n = 29; 32.2%). The majority of studies were conducted in the USA or Canada (n = 50; 55.6%), Asian countries (n = 15; 16.7%) or European countries (n = 10; 11.1%). Most studies reported on a population of individuals with mixed ethnicity (n = 53; 58.9%) and in urban settings (n = 83; 92.2%). Characteristics of all included studies have been summarized in Table 1 and individually described in S3 Table and per region in S4 Table.

### PrEP retention

S5 Table displays retention outcomes for all included studies per pre-defined time point. Definitions of retention were predominately based on follow-up (n = 56; 62.9%). Definitions related to continuation were used less frequently (n = 16; 18.0%), and found in n = 7/47 (14.9%) studies with daily PrEP, n = 0/1 (0%) with event-driven PrEP, n = 2/5 (40.0%) with mixed daily and event-driven PrEP, and n = 7/35 (20.0%) with unspecified PrEP modality.

Fig 2A depicts the observed cumulative probability of PrEP retention for each study and timepoint, alongside the predicted probability from the random-effects survival model. The model tends to give lower estimates of this probability than observed, particularly at later timepoints. From the model, the pooled cumulative probability of PrEP retention was 77.0% (95%CI = 73.4–80.2%) at 6 months, 64.7% (95%CI = 57.6–70.9%) at 12 months, 48.5% (95%CI = 36.4–59.4%) at 24 months, and 24.1% (95%CI = 10.3–41.2%) at 60 months.

Sensitivity analysis demonstrated that the pooled cumulative probability of PrEP retention was higher when including only randomized control trials (Fig 2B), relatively unchanged for only studies with more than 100 participants (Fig 2C), and mostly unchanged for only studies with a risk score bias of less than 5 (Fig 2D). For randomized control trials only, the pooled cumulative probability of PrEP retention was 77.3% (95%CI = 68.9–83.6%) at 6 months, 71.1% (95%CI = 56.7–81.5%) at 12 months, and 63.8% (95%CI = 40.1–80.2%) at 24 months. There were no other studies beyond this time

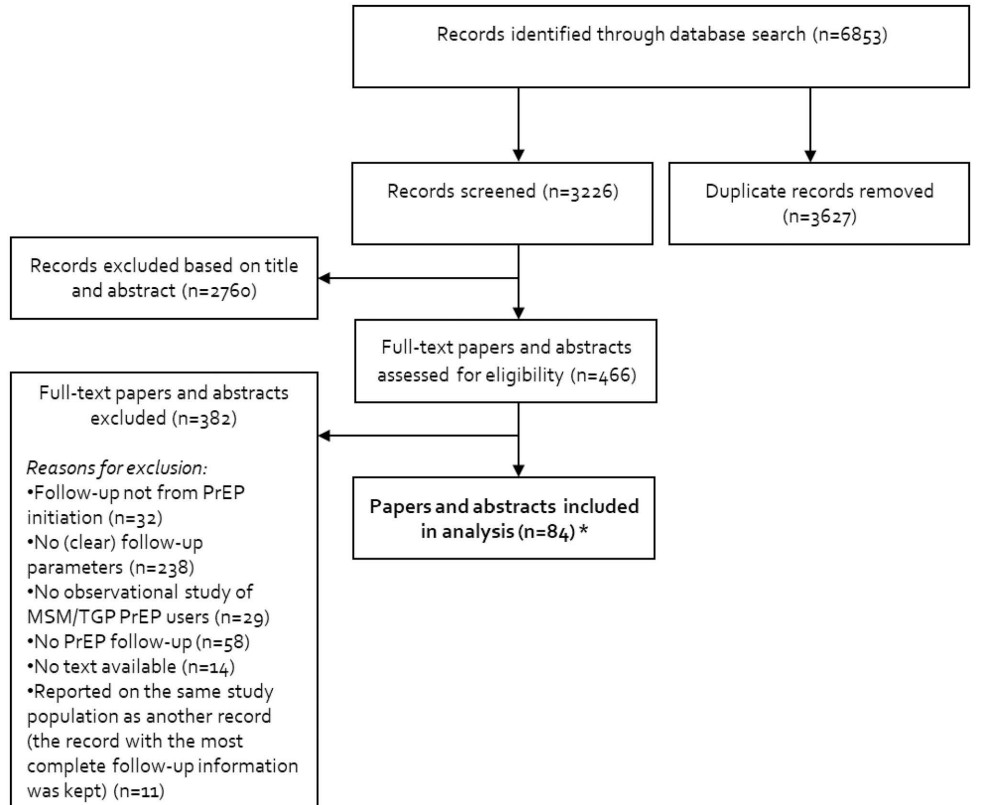

**Fig 1. Flow diagram of study search and inclusion.** *Studies with two study arms (n = 6) were included as one study but were analyzed separately. In total, the number of inclusions is 84 individual studies, amounting to 90 inclusions (i.e., study cohorts) in analysis. Abbreviations: MSM = men who have sex with men; TGP = transgender person; PrEP = pre-exposure prophylaxis.

point. For studies with more than 100 participants only, the pooled cumulative probability of PrEP retention was 79.0% (95%CI = 75.1–82.4%) at 6 months, 67.6% (95%CI = 60.0–74.1%) at 12 months, 52.1% (95%CI = 38.9–63.8%) at 24 months, and 27.9% (95%CI = 11.6–47.0%) at 60 months. For studies with a risk score bias of more than 5 only, the pooled cumulative probability of PrEP retention was 76.3% (95%CI = 72.4–79.8%) at 6 months, 62.8% (95%CI = 55.1–69.6%) at 12 months, 45.0% (95%CI = 32.4–56.8%) at 24 months, and 19.4% (95%CI = 7.0–36.5%) at 60 months.

The model also produced similar results when only restricting data collected in the first 24 months (Fig 2E). Using these data, the pooled cumulative probability of PrEP retention was 76.9% (95%CI = 73.2–80.1%) at 6 months, 64.6% (95%CI = 57.4–70.9%) at 12 months, 48.4% (95%CI = 36.1–59.6%) at 24 months.

## Determinants of variability in loss of PrEP retention

Table 2 summarized the results of the meta-regression analysis. Studies conducted in Europe (*p* < 0.001), Australia (*p* < 0.001) and multiple regions (*p* = 0.003) compared to North America, and studies with follow-up intervals of more than three months (*p* = 0.028) compared to those with follow-up visits every 3 months or less had loss of PrEP retention rates that were significantly lower (i.e., increased retention). In contrast, studies conducted in Africa (vs. North America; *p* = 0.005), studies from LMIC (vs. HIC, *p* = 0.035), studies with unspecified ethnicity (vs. mixed ethnicity; *p* = 0.031), unspecified PrEP regimen (vs. daily; *p* < 0.001), free-of-charge STI testing (vs. unspecified; *p* = 0.007) and in cohorts reported in abstracts (vs. peer-reviewed papers; *p* < 0.001) had significantly higher loss of PrEP retention rates. Studies

**Table 1. Description of study characteristics.**

| Variables | n/N (%) or median [IQR] |
|---|---|
| **Region** | |
| North America | 53/90 (58.9) |
| Europe | 5/90 (5.6) |
| Australia | 6/90 (6.7) |
| South America | 6/90 (6.7) |
| Africa | 5/90 (5.6) |
| Asia | 12/90 (13.3) |
| Multiple | 3/90 (3.3) |
| Country income setting | |
| High income | 64/90 (71.1) |
| Low to middle income | 23/90 (25.6) |
| Multiple | 3/90 (3.3) |
| **Calendar year** | |
| Year begun [N = 82]* | 2015 [2014-2017] |
| Year completed [N = 82]† | 2017 [2016-2018] |
| Year published [N = 90] | 2019 [2018-2020] |
| **Study population** | |
| MSM (>80%) | 72/90 (80.0) |
| MSM, TGP and other | 18/90 (20.0) |
| **Percentage of study population (%)** ‡ | |
| MSM [N = 82] | 98.5 [91.7-100] |
| TGP [N = 82] | 4.4 [1.0-9.8] |
| Other [N = 82] | 5.0 [1.3-10.5] |
| **Geographic setting** | |
| Urban | 83/90 (92.2) |
| Urban/rural or rural only | 7/90 (7.8) |
| **Ethnicity** | |
| Mixed | 53/90 (58.9) |
| Black only | 6/90 (6.7) |
| Asian only | 3/90 (3.3) |
| Unspecified | 28/90 (31.1) |
| **Median of study median age (years)** [N = 34] | 32 [25-35] |
| **Frequency of follow-up visits** | |
| <3 months | 72/87 (82.8) |
| ≥3 months | 15/87 (17.2) |
| **Retention terminology**¶ | |
| Continuation | 16/89 (18.0) |
| Follow-up | 56/89 (62.9) |
| Mixed/Other | 17/89 (19.1) |
| **Publication type** | |
| Paper | 58/90 (64.4) |
| Abstract | 32/90 (35.6) |
| **PrEP regimen** | |
| Daily | 47/90 (52.2) |
| Event-Driven | 1/90 (1.1) |
| Both | 6/90 (6.7) |

*(Continued)*

**Table 1.** (Continued)

| Variables | n/N (%) or median [IQR] |
|---|---|
| Unspecified | 36/90 (40.0) |
| **PrEP cost** | |
| Completely free | 25/90 (27.8) |
| Not entirely free | 10/90 (11.1) |
| Unspecified | 55/90 (61.1) |
| **STI testing cost** | |
| Not entirely free | 1/90 (1.1) |
| Completely free | 6/90 (6.7) |
| Unspecified | 83/90 (92.2) |

Data retrieved from 84 studies published between 2010 and 2021. All continuous variables are summarized as median (IQR). Abbreviations: IQR = inter-quartile range; MSM = men who have sex with men; TGP = transgender person; PrEP = pre-exposure prophylaxis; STI = sexually transmitted infection.

*Defined as the year in which the study with PrEP follow-up was initiated.

†Defined as the year in which the study with PrEP follow-up was stopped.

¶Defined as the terminology used in the study to indicate or determine PrEP retention.

with more recent study start dates ($p=0.015$), and with larger proportions of participants other than MSM or TGP ($p=0.029$) had higher loss of PrEP retention rates; whereas higher median age of participants was associated with decreased loss of retention rates ($p=0.035$). There was no evidence that retention definition, either urban or rural geographic setting, or PrEP cost were associated with differences in loss of PrEP retention. There was also no evidence of interaction between year of study start and any of the factors.

Multivariable analysis was precluded by the fact that certain levels of factors were commonly shared within studies.

### Quality assessment

The risk of bias assessment shows that there was a low overall risk of bias (S2 Table). Heterogeneity was consistently high between-studies from data reported throughout the first year of follow-up (month 3, $I^2=99.2\%$; month 6, $I^2=99.3\%$; month 9, $I^2=99.3\%$; month 12, $I^2=99.6\%$). Accordingly, between-study heterogeneity was significant ($p<0.0001$ for all time points). GOSH plots are provided in S1 Fig, demonstrating no clear combinations of studies that contributed to heterogeneity. From these analyses, the lowest attainable $I^2$ was 89.1% at month 3, 92.5% at month 6, 89.4% at month 9, and 97.0% at month 12. There were no influential studies identified at any of the timepoints (data not shown). Additionally, significant publication bias was observed at each of these timepoints (month 3, $p<0.0001$; month 6, $p<0.0001$; month 9, $p=0.0005$; month 12, $p<0.0001$; funnel plots provided in S2 Fig), with larger studies showing higher cumulative probabilities of PrEP retention.

### Discussion

This systematic review and meta-analysis estimated a 77.0% cumulative probability of PrEP retention at 6 months after initiation of oral PrEP. This probability declined to 48.5% at 24 months and to 24.1% at 60 months. Along with the substantial heterogeneity in this probability observed across studies, and the multiple factors associated with loss of PrEP retention rates, our findings highlight that continuing oral PrEP use is challenging for the majority of those who start PrEP.

It is important to consider that retention is defined in different ways across studies, but differences in definitions seem to have little effect on loss of retention rates. For daily PrEP, continuation of PrEP intake could be an appropriate measure of retention. However, PrEP discontinuation is not *per se* an undesirable endpoint. If individuals have a low probability or low perceived likelihood of acquiring HIV, discontinuation is warranted with appropriate counselling [21]. Research has shown

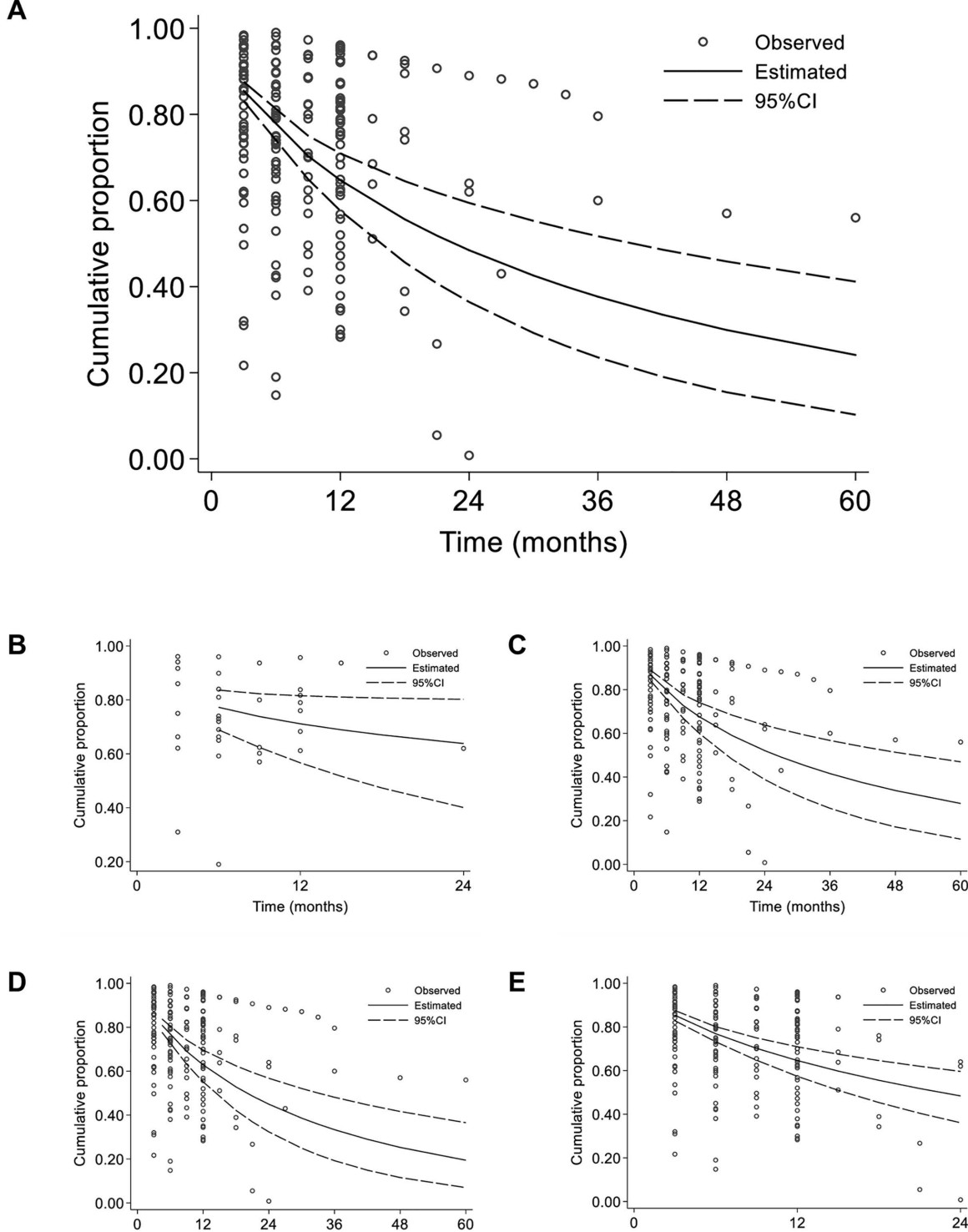

**Fig 2. Cumulative probability of individuals retained on pre-exposure prophylaxis for preventing HIV over time.** Data retrieved from 90 studies conducted between 2010 and 2020 and ranging from 3 to 60 months of follow-up (median = 12) **(A)**. Individual values for studies are depicted as grey dots. Sensitivity analysis are also provided when including only randomized control trials **(B)**, only studies with more than 100 participants **(C)**, only studies with a risk score bias of less than 5 **(D)**, and while restricting data collected in the first 24 months **(E)**.

**Table 2. Meta-regression for relative differences in loss of PrEP retention rates.**

| Variables | Univariable model | | |
|---|---|---|---|
| | *HR* | *(95% CI)* | *P* |
| **Region** | | | |
| North America | Ref | | |
| Europe | 0.31 | (0.15-0.62) | 0.001 |
| Australia | 0.30 | (0.17-0.56) | <0.001 |
| South America | 0.95 | (0.54-1.68) | 0.87 |
| Africa | 2.47 | (1.32-4.62) | 0.005 |
| Asia | 1.00 | (0.64-1.58) | 0.99 |
| Multiple | 0.36 | (0.18-0.71) | 0.003 |
| Country income setting | | | |
| High income | Ref | | |
| Low to middle income | 1.46 | (1.03-2.09) | 0.035 |
| Multiple | 0.43 | (0.21-0.89) | 0.023 |
| **Calendar year (per year)** | | | |
| Year begun* | 1.11 | (1.02-1.20) | 0.015 |
| Year completed† | 1.08 | (0.98-1.20) | 0.13 |
| Year published | 1.11 | (0.98-1.27) | 0.10 |
| **Study population** | | | |
| MSM (>80%) | Ref | | |
| MSM, TGP and other | 1.14 | (0.78-1.66) | 0.51 |
| **Percentage of study population (per %)‡** | | | |
| MSM | 0.98 | (0.97-1.00) | 0.053 |
| TGP | 1.02 | (0.99-1.04) | 0.11 |
| Other | 1.03 | (1.00-1.06) | 0.029 |
| **Geographic setting** | | | |
| Urban | Ref | | |
| Urban and rural or rural only | 1.15 | (0.69-1.91) | 0.60 |
| **Ethnicity** | | | |
| Mixed | Ref | | |
| Black only | 0.84 | (0.45-1.57) | 0.59 |
| Asian only | 0.85 | (0.28-3.60) | 0.77 |
| Unspecified | 1.47 | (1.04-2.08) | 0.031 |
| **Median of study median age (per year)‖** | 0.96 | (0.92-0.99) | 0.035 |
| **Frequency of follow-up visits** | | | |
| <3 months | Ref | | |
| ≥3 months | 0.58 | (0.36-0.94) | 0.028 |
| **Outcome definition¶** | | | |
| Continuation | Ref | | |
| Follow-up | 1.06 | (0.70-1.61) | 0.78 |
| Mixed/Other | 0.65 | (0.39-1.07) | 0.091 |
| **Study type** | | | |
| Paper | Ref | | |
| Abstract | 1.94 | (1.40-2.67) | <0.001 |
| **PrEP regimen** | | | |
| Daily | Ref | | |
| Event-Driven | 0.35 | (0.10-1.27) | 0.11 |

*(Continued)*

**Table 2.** (Continued)

| Variables | Univariable model | | |
|---|---|---|---|
| | *HR* | *(95% CI)* | *P* |
| Both | 0.95 | (0.53-1.71) | 0.86 |
| Unspecified | 2.06 | (1.53-2.79) | <0.001 |
| **PrEP cost** | | | |
| Completely free | Ref | | |
| Not entirely free | 1.50 | (0.78-2.87) | 0.23 |
| Unspecified | 1.51 | (0.97-2.36) | 0.070 |
| **STI cost** | | | |
| Not entirely free | 4.45 | (1.50-13.17) | 0.007 |
| Completely free | 0.93 | (0.51-1.68) | 0.80 |
| Unspecified | Ref | | |

Meta-regression performed on data retrieved from 90 study cohorts conducted between 2010 and 2021. This model was run for each of the timepoints at which the majority of studies reported data (i.e., months 3, 6, 9, and 12).

Abbreviations: MSM = men who have sex with men; TGP = transgender person; PrEP = pre-exposure prophylaxis; STI = sexually transmitted infection.

*Defined as the year in which the study with PrEP follow-up was initiated; data missing for 8 studies.

†Defined as the year in which the study with PrEP follow-up was stopped; data missing for 8 studies.

‡Missing data or nonapplicable study population in 9 (MSM), 46 (TGP) and 65 (other) studies.

||Missing data in 56 studies.

¶Defined as the terminology used in the study to indicate or determine PrEP retention.

that those who discontinue continuous PrEP as they are no longer engaging in behaviours associated with HIV acquisition may be more inclined to re-initiate when their likelihood of HIV increases again [11,22]. This type of dosing would essentially be event-driven PrEP, which would require other measures of retention, such as those based on follow-up. Retention in PrEP care could be particularly helpful as timely access to PrEP and counselling after discontinuation are essential to ensure HIV prevention [21].

Although MSM and TGP are key target populations for PrEP use, other populations (e.g., women, heterosexuals, people who inject drugs) engaging in behaviors or belonging to networks which HIV acquisition is more common are less likely to be on PrEP, regardless of need [23]. Even though we included studies that were predominately MSM and TGP, those including a larger proportion of participants outside these groups, most of which were unspecified, had higher loss of PrEP retention rates. This may be attributable to PrEP care not being tailored to non-MSM groups [7,24], and the fact that the perceived likelihood of acquiring HIV could vary in these groups and perceptions might not always match their behavioral vulnerability to HIV [25]. Nevertheless, many studies had missing information concerning population composition, making it difficult to link the lower retention levels to any specific subgroup. Further meta-analyses should strive to include studies in key populations other than MSM and TGP as more data become available.

Another population-specific factor related to loss of PrEP retention rates was age. We found that with older age, these rates were lower. This finding mirrors other outcomes specific to oral PrEP use, such as more frequent switching between daily and event-driven regimens [26] and poorer adherence [27] in younger individuals. Moreover, younger MSM have often been found to experience barriers to PrEP care as a result of limited access, high costs, stigma, and perceived lower likelihood of HIV despite behavioral vulnerability [28]. These issues could partly explain the lower levels of PrEP retention found among younger MSM [29,30].

Furthermore, there were regional differences in loss of PrEP retention rates. When compared to North America, studies in Europe and Australia had lower loss of retention rates, while those in Africa had higher rates. When dividing these

countries based on income, loss of PrEP retention was more commonly observed in LMIC settings. Regional differences are difficult to explain, but could be due to population differences [31] and variations in levels of PrEP availability or the ability of healthcare systems to provide and support PrEP use and adherence [32]. For instance, across Europe [33] and in Australia [34], multiple large-scale PrEP programs have been initiated, allowing for more affordable and easily accessible PrEP. In contrast, insurance-related and social stigma barriers observed in North America have been known to negatively impact PrEP use [23]. Moreover, research shows that low perceived likelihood of acquiring HIV, experienced stigma and difficulties in returning to PrEP care appointments, often experienced by MSM and TGP in African countries, could hinder oral PrEP use [35–37]. The factors affecting oral PrEP use that were identified in these studies could have downstream consequences for PrEP retention, as evidenced in our meta-analysis. The observed regional differences could also be the result of differences in study populations and design; given that Asian studies included a lower proportion of MSM and more frequent visits were commonly found in North American, South American and African studies (S4 Table).

Loss of PrEP retention rates were found to be significantly lower among studies with ≥3 monthly follow-up visits (vs. <3 monthly), suggesting that those who are required to engage in more intense PrEP care are less likely to be retained on PrEP. The experienced burden of frequent clinic visits [11] or changing sexual behavior over time are likely reasons for this finding. The effect of increased monitoring on PrEP adherence has rarely been quantitatively studied, but has surfaced as a prominent issue in several qualitative studies [38,39]. Earlier study start dates were also associated with loss of PrEP retention rates that were lower (i.e., increased retention). For these earlier studies (i.e., before 2016), which were more often demonstration or implementation projects, included individuals may have been more engaged and motivated to use PrEP, as formal PrEP access was limited [24,40]. Balance might be needed between the disadvantages (e.g., burden of clinical visits, lower proportion of retention) and benefits (e.g., earlier diagnosis and treatment of STI and HIV) of more frequent PrEP visits.

One curious finding is the association between the unspecified levels of several factors and higher loss of PrEP retention rates. For instance, these rates were higher when studies did not specify PrEP regimen (vs. daily regimen) and when studies did not specify ethnicity (vs. mixed ethnicity). Only one study involved solely event-driven PrEP, which makes inference on this PrEP modality challenging. Roughly 40% of the included studies did not report PrEP modality and if these studies did include those undergoing event-driven PrEP, it would have helped provide a clearer understanding of PrEP retention for these users, Previous studies have shown that racial disparities exist in PrEP retention [41]; as such, understanding what constitutes "unspecified" for these variables would appear valuable in identifying population- and program-specific factors influencing retention on PrEP. However, as we have observed in this meta-analysis, large-scale and long-term studies generally do not measure the impact of such factors on PrEP retention. This gap in knowledge could be due to the lack of collection of this data, and hence more consistency in the collection and reporting of information is needed when evaluating PrEP programs.

Strengths of our systematic review and meta-analysis are the large number of studies, long follow-up times, and the fact that we did not solely assess one aspect across studies (i.e., population-, program-, and study-specific characteristics were studied). In addition, we included studies reporting on TGP, an underrepresented population in research. This meta-analysis attempts to provide a much-needed summary of PrEP retention outcomes in the context of TGP groups; however, we acknowledge that this remains a prominent disparity in PrEP-related research.

There are some limitations of our study. First, publication bias was pervasive across studies; we therefore need to be aware of a potential overestimation of PrEP retention compared to routine practice [42]. Second, we only considered loss of PrEP retention as an outcome and assumed that those no longer retained never re-initiated PrEP. A relatively large proportion of individuals are known to re-initiate PrEP after discontinuation [7], but the variation in definitions of PrEP retention across studies makes it difficult to distinguish the underlying reasons for loss of PrEP retention. Third, we did not search the Cochrane or Embase databases due to access restrictions and did not include conferences such as the HIV Research for Prevention Conference; hence, although there is likely overlap across these databases/

conferences, some studies may have been missed. Fourth, because certain levels of factors were commonly shared between studies, a multivariable meta-regression survival model could not be constructed. The model also tended to give lower estimates of the probability of remaining on PrEP than observed, particularly at later timepoints. Fifth, the vast majority of published studies represented European, Australian and North American settings. Although some studies from South East Asia, Africa and South America were included, the pooled estimates might not be fully representative of these and other non-included regions. Sixth, this systematic review included a large number of studies, yet a relatively limited number of studies were found to report long-term outcomes after month 24. Finally, we only included studies using oral PrEP, meanwhile injectable cabotegravir can also be used to effectively prevent HIV acquisition [43]. In mid 2022, cabotegravir PrEP has been recommended as an option for individuals at very high risk of HIV acquisition [44], but many HIV prevention programs seem to be struggling with implementing cabotegravir PrEP [45]. Further research on PrEP retention would be needed for the use of cabotegravir PrEP, particularly when more observational data become available.

Our findings could assist in improving PrEP care as it is crucial that healthcare professionals engage and/or re-engage PrEP users by critically evaluating PrEP need and identifying barriers. Engagement in PrEP care should include informing and counseling by providing information on retention benefits or alternative methods for HIV prevention, and the provision of multiple pathways back to PrEP care [46]. Moreover, our findings might warrant tailored PrEP care (e.g., less frequent visits, free-of-charge PrEP and STI testing, or online PrEP care), considering the variability in PrEP retention across different population characteristics, such as ethnicity, age and being non-MSM/TGP. We must be aware that different populations may experience different barriers to PrEP care [31], which could lead to loss of retention in varying ways.

In conclusion, PrEP retention became markedly lower with longer time on oral PrEP and differed between populations and across program-specific characteristics. Studies must provide consistent information on what constitutes PrEP retention for clear comparisons across studies, which could provide a more precise understanding of the structural and individual factors influencing the variability in this outcome. PrEP programs with more intense monitoring schedules or requiring that patients pay for some of their PrEP/STI testing services may face lower rates of retention; however, clearer evidence that these program factors do indeed have an effect on retention is needed. The heterogeneity found in the cumulative probability retained on PrEP across study cohorts highlights the need for more tailored PrEP retention strategies.

## Supporting information

**S1 Table. Search strategy.**
(DOCX)

**S2 Table. Risk of bias assessment using the Newcastle-Ottawa Scale (NOS) for assessing the quality of nonrandomized studies in this systematic review and meta-analyses.** This table also includes the citations for every study; see S1 References for reference list. †We evaluated comparability on the basis of the analysis and not the design (as studies were single-arm). This evaluation was carried out as follows: whether proportions were adjusted for age and (if diverse populations were included) gender (one star), and whether proportions were additionally adjusted for other factors (one star). The comparability criteria were applied similarly for randomized control trials and observational studies. ‡Randomized control trials were included in the systematic review as separate cohort studies and thus the Newcastle-Ottawa Quality Assessment Scale for cohort studies was applied hereto.
(DOCX)

**S3 Table. Characteristics of included studies.** Abbreviations: MSM = men who have sex with men; TGP = transgender person; PrEP = pre-exposure prophylaxis; IQR = interquartile range; N.A. = not applicable because missing or not reported.
(DOCX)

**S4 Table. Description of study characteristics according to region of study.** Data retrieved from 90 studies conducted between 2010 and 2020; three studies conducted in multiple geographic regions were not included in this analysis. All continuous variables are summarized as median [IQR]. Abbreviations: IQR = interquartile range; MSM = men who have sex with men; TGP = transgender person; PrEP = pre-exposure prophylaxis; STI = sexually transmitted infection. *Defined as the year in which the study with PrEP follow-up was initiated; data missing for 8 studies: North America (n = 4), Australia (n = 1), South America (n = 1), Africa (n = 2). †Defined as the year of last reported follow-up in the study; data missing for 8 studies: North America (n = 4), Australia (n = 1), South America (n = 1), Africa (n = 2). ‡Missing data or nonapplicable study population in 8 studies for MSM [North America (n = 6), South America (n = 2)], 45 for TGP [North America (n = 30), Europe (n = 1), Australia (n = 3), South America (n = 3), Africa (n = 4), Asia (n = 4)] and 63 for other populations [North America (n = 35), Europe (n = 4), Australia (n = 3), South America (n = 6), Africa (n = 5), Asia (n = 10)]. ‖Missing data in 54 studies: North America (n = 37), Australia (n = 2), South America (n = 5), Africa (n = 4), Asia (n = 6). ¶Defined as the terminology used in the study to indicate or determine PrEP retention. $Only one study with reported data.
(DOCX)

**S5 Table. Table of primary retention outcomes.** Abbreviations: N.A. = not applicable in case of missing/non-reported data.
(DOCX)

**S1 Fig. GOSH plots assessing combinations of studies contributing to heterogeneity on PrEP retention.** Observed outcome (i.e., ln(-ln) transformed probability of PrEP retention) and the $I^2$ heterogeneity statistic are plotted from the 10,000 randomly selected study subsets of various sizes at month 3 (A), month 6 (B), month 9 (C), and month 12 (D) after PrEP initiation. No patterns causing heterogeneity emerged. The smoothed histograms of both variables are provided alongside the plot.
(TIF)

**S2 Fig. Funnel plots assessing publication bias in studies on PrEP retention.** Observed outcome (i.e., ln(-ln) transformed probability of PrEP retention) and standard errors corresponding to ln(-ln) transformed probabilities are plotted at month 3 (A), month 6 (B), month 9 (C), and month 12 (D) after PrEP initiation. Asymmetry in the plots was tested using Egger's test for bias. At each timepoint, it was apparent that larger studies (i.e., those with lower standard errors) exhibited higher cumulative probabilities of PrEP retention. This bias was significant at each timepoint (month 3, $p < 0.0001$; month 6, $p < 0.0001$; month 9, $p = 0.0005$; month 12, $p < 0.0001$).
(TIF)

**S1 References.** References of included papers and abstracts (cited in S2 Table).
(DOCX)

**S1 Appendix. PRISMA checklist.**
(PDF)

## Author contributions

**Conceptualization:** David O. T. ten Hoff, Maria Prins, Liza Coyer, Anders Boyd.

**Data curation:** Feline de la Court, David O. T. ten Hoff, Liza Coyer, Anders Boyd.

**Formal analysis:** Anders Boyd.

**Funding acquisition:** Maria Prins.

**Investigation:** Feline de la Court, Liza Coyer, Anders Boyd.

**Methodology:** Liza Coyer, Anders Boyd.

**Project administration:** Feline de la Court, Maria Prins.

**Resources:** Maria Prins.

**Supervision:** Maria Prins, Anders Boyd.

**Visualization:** Feline de la Court.

**Writing – original draft:** Feline de la Court, Liza Coyer, Anders Boyd.

**Writing – review & editing:** Feline de la Court, Maria Prins, Elske Hoornenborg, Maarten F. Schim van der Loeff, Liza Coyer, Anders Boyd.

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
