## [Decision Letter · Decision Letter 0]

6 Dec 2023

PONE-D-23-22373Long-term pre-exposure prophylaxis retention among men who have sex with men and transgender persons: systematic review and meta-analysisPLOS ONE

Dear Dr. de la Court,

Thank you for submitting your manuscript to PLOS ONE. After careful consideration, we feel that it has merit but does not fully meet PLOS ONE’s publication criteria as it currently stands. Therefore, we invite you to submit a revised version of the manuscript that addresses the points raised during the review process.

We look forward to receiving your revised manuscript.

Kind regards,

Septime P.H. Hessou, M.D.

Academic Editor

PLOS ONE

Journal Requirements:

"This study is part of the OptiPrEP (optimizing pre-exposure prophylaxis roll-out among men having sex with men) project, funded by the Aidsfonds (P-54601). "

"MP obtained unrestricted research grants and speaker/ advisory fees from Gilead Sciences, Abbvie and MSD; all of which were paid to her institute and were unrelated to the current work. EH obtained unrestricted research grants from Gilead Sciences, which were paid to her institute and were unrelated to the current work. The institution of M. F. Schim van der Loeff receives study funding from GSK; he served on advisory boards of GSK and Merck/MSD; fees were paid to his institution. The other authors report no conflicts of interest.

The current study is part of the OptiPrEP (optimizing pre-exposure prophylaxis roll-out among men having sex with men) project, funded by the Aidsfonds (P-54601).

"MP obtained unrestricted research grants and speaker/ advisory fees from Gilead Sciences, Abbvie and MSD; all of which were paid to her institute and were unrelated to the current work. EH obtained unrestricted research grants from Gilead Sciences, which were paid to her institute and were unrelated to the current work. The institution of M. F. Schim van der Loeff receives study funding from GSK; he served on advisory boards of GSK and Merck/MSD; fees were paid to his institution.

The authors have declared that no competing interests exist"

Please confirm that this does not alter your adherence to all PLOS ONE policies on sharing data and materials, by including the following statement: "This does not alter our adherence to  PLOS ONE policies on sharing data and materials.” (as detailed online in our guide for authors http://journals.plos.org/plosone/s/competing-interests ). If there are restrictions on sharing of data and/or materials, please state these. Please note that we cannot proceed with consideration of your article until this information has been declared.

4. In your Data Availability statement, you have not specified where the minimal data set underlying the results described in your manuscript can be found. PLOS defines a study's minimal data set as the underlying data used to reach the conclusions drawn in the manuscript and any additional data required to replicate the reported study findings in their entirety. All PLOS journals require that the minimal data set be made fully available. For more information about our data policy, please see http://journals.plos.org/plosone/s/data-availability .

Upon re-submitting your revised manuscript, please upload your study’s minimal underlying data set as either Supporting Information files or to a stable, public repository and include the relevant URLs, DOIs, or accession numbers within your revised cover letter. For a list of acceptable repositories, please see http://journals.plos.org/plosone/s/data-availability#loc-recommended-repositories . Any potentially identifying patient information must be fully anonymized.

Important: If there are ethical or legal restrictions to sharing your data publicly, please explain these restrictions in detail. Please see our guidelines for more information on what we consider unacceptable restrictions to publicly sharing data: http://journals.plos.org/plosone/s/data-availability#loc-unacceptable-data-access-restrictions . Note that it is not acceptable for the authors to be the sole named individuals responsible for ensuring data access.

Additional Editor Comments :

The acceptance and retention of people on PrEP are intimately linked to the high level of HIV prevalence and/or incidence in countries and at the level of the people, in addition to socio-cultural norms in Africa and elsewhere, the level of health development and the level of education of the people.

It would be wise to clearly define in the methodology what long-term retention means in terms of procedure, type of PrEP and timeframe. Especially as PrEP must be continuous at all times to avoid any risk .......

Introduction

At this level, it would be good to already hypothesize about program characteristics that may influence target retention on PrEP,

The absence of this made it difficult to understand your results and conclusions, as this was the aim of the study.

Methods

- Definition PICOS

In this methodological part, as you have used PSRISMA, please present clearly : PICOS

P= Populations

I= Interventions

C=comparator

O= outcome

S= Study

- Long-term PrEP retention

It would be wise to clearly define in the methodology what long-term retention means in terms of procedure, type of PrEP and timeframe. Especially as PrEP must be continuous at all times to avoid any risk .......

Data Extraction

How is it possible to do a follow-up or retention study with on-demand or event-related PreP?

Statistical analysis

Statistical approach not easy but well adapted and well developed and can lead to good results and conclusions

Results

For RCTs, how was the analysis done if the arm does not contain the drug, is it ethical please?

Discussion

Very good and detailed discussion of the real problems of bias in this study.

The problem posed by the discussion is that of the completeness of the studies and consequently of the extrapolation of the results: the majority of the studies are from North America, just 5% for each of the other parts of the world, so the discussion of this representativeness bias needs to be strengthened.

Conclusion

Reinforce the conclusion with program factors that actually influence MSM and TG retention on PrEP

This study poses an important problem that other studies will be able to address: the relationship between the time taken to maintain or retain on PrEP and the risk of HIV infection.

Reviewers' comments:

Reviewer's Responses to Questions

**Comments to the Author**

1. Is the manuscript technically sound, and do the data support the conclusions?

Reviewer #1: Partly

Reviewer #2: Partly

Reviewer #3: Yes

2. Has the statistical analysis been performed appropriately and rigorously? 

Reviewer #1: I Don't Know

Reviewer #2: No

Reviewer #3: I Don't Know

3. Have the authors made all data underlying the findings in their manuscript fully available?

Reviewer #1: Yes

Reviewer #2: Yes

Reviewer #3: Yes

4. Is the manuscript presented in an intelligible fashion and written in standard English?

Reviewer #1: No

Reviewer #2: Yes

Reviewer #3: Yes

5. Review Comments to the Author

Reviewer #1: Major revision:

It includes data on event driven PrEP but it is unclear how many participants were receiving event driven PrEP. For about 40% this is unspecified. Retention in event driven PrEP is a slight misnomer. And the paper does not include different definitions of retention for event driven and daily PrEP which I think it warranted.

Minor revisions:

- Suggest including details about daily vs event driven PrEP in the intro

- Is there information missing from table 1? I cannot tell how many studies in each of the calendar year brackets

- In table 2 - I am unclear how to interpret the rows on percentage of study population (%). Can this be better displayed and explained in the text?

- Sometimes the sentence structure is confusing to me. For example "PrEP discontinuation was significantly

lower as median age of participants was lower (p=0.035)." Clearer to say: Younger median age of participants was associated with continuing on PrEP. Although this seems at odds with this statement in the discussion: "We found

that with older age, the rates of PrEP discontinuation were lower." I think the whole paper could do with an edit.

- Unclear why the defintion of outcome was included as a variable. I cannot understand why definition would impact outcome and it is not well explained. I suggest excluding this from the paper.

Reviewer #2: The authors performed a systematic review and meta-analysis of 84 studies that reported on retention of oral PrEP for HIV and included predominantly MSM and TGP retrieved from the PubMed and Ovid online databases, capturing demonstration projects or observational studies published over the period from January 1, 2010 to March 24, 2021. Both observational and randomised studies were included. The medline research appears to be accurate, follows the PRISMA guidelines and the paper is well written but I have some general concerns.

Main Points

1. The search has not been updated since end of March 2021. Many more studied could have published since that date that could be included here.

2. Aa a corollary to point #1, authors could have investigated whether retention rates have changed in recent years after the introduction of long-lasting regimens for PreP.

3. According to the 'star system' for the Newcastle-Ottawa Quality Assessment Scale studies, among other things, are judged on the comparability of the groups. The meaning of using this system to judge RCTs in unclear as by definition groups are comparable because of randomisation.

4. Because of possible publication bias and potential confounding I would perform a sensitivity analysis after restricting only to the RCTs. Similarly, I would like to see the results of a sensitivity analysis after excluding studies with small sample size.

5. Only few studies reported the proportion of retained care after month 24. I suggest that predictions should be given up to month 24 because after that these are based on little data. This also explains why the proportion of retained in care is largely underestimated by the model compared to observed at month 60.

6. Table S2 shows that there were studies with risk score bias of 5. I would like to see the results of a sensitivity analysis after excluding these.

7. There is significant heterogeneity between the studies which could have affected the overall estimate of the retention in care. Authors should consider using the sequential and combinatorial algorithms developed by Patsopoulos et al. [Patsopoulos NA, Evangelou E, Ioannidis JPA. Sensitivity of between-study heterogeneity in meta-analysis: proposed metrics and empirical evaluation. Int J Epidemiol 2008;37:1148−57] which evaluate the change in between-study heterogeneity as one or more studies are excluded from the calculations. The proposed algorithms can be routinely applied in meta-analyses as standardized sensitivity analyses for heterogeneity, and they can either generate the number of studies that have to be omitted to decrease I2 below the specified threshold, or estimate the corresponding proportion of studies.

8. Despite the statistical significance some of the effects in the meta-regression analysis (e.g heterosexuals/women/PWID vs. MSM was only 3% difference) are of dubious clinical significance. In contrast, there may be up to a 2.9-fold increase in discontinuation when PreP was not free but the meta-regression is underpowered to detect this difference. I would bettwe qualify these findings in the Discussion section.

Other points

1. Line 252 – looks like there is a typo. PrEP discontinuation was significantly lower as median age of participants was OLDER (not lower) (p=0.035). There is a 4% lower risk of discontinuation per 1 year older. It does make sense that the younger the participants the higher the rates of discontinuation, not vice versa (as discussed)

Reviewer #3: The authors present the findings from a very relevant and well-conducted systematic review with a meta-analysis of long-term retention on pre-exposure prophylaxis. I have no major comments on the manuscript, it is well-written, and found the methodology rigorous and technically sound. I congratulate the authors on this.

My minor comments are:

- The authors refer in the abstract that they will estimate the extent to which variability in PrEP retention is attributable to population- and program-specific characteristics. However, variables such as the type of publication are used. Please, provide the rationale for such inclusion. Additionally, I wondered whether in the objective the authors should use “associated” instead of “attributable”.

- In the results section I would suggest presenting the HR instead of just the p-values which are more informative about the association.

- Would it be worth discussing if some determinants were more important in earlier or later studies? I would imagine that both the population and program characteristics may have differed according to the year of study beginning and so the determinants may have also differed.

6. PLOS authors have the option to publish the peer review history of their article (what does this mean? ). If published, this will include your full peer review and any attached files.

**Do you want your identity to be public for this peer review?** For information about this choice, including consent withdrawal, please see our Privacy Policy .

Reviewer #1: No

Reviewer #2: No

Reviewer #3: No

---

## [Author Response · Author response to Decision Letter 1]

19 Feb 2024

Please see our submitted Response to Reviewers document for our response to the reviewers and editor.

---

## [Decision Letter · Decision Letter 1]

27 Jan 2025

PONE-D-23-22373R1Oral pre-exposure prophylaxis retention among men who have sex with men and transgender persons: systematic review and meta-analysisPLOS ONE

Dear Dr. de la Court,

Thank you for submitting your manuscript to PLOS ONE. After careful consideration, we feel that it has merit but does not fully meet PLOS ONE’s publication criteria as it currently stands. Therefore, we invite you to submit a revised version of the manuscript that addresses the points raised during the review process.

We look forward to receiving your revised manuscript.

Kind regards,

Emma Campbell, Ph.D

Staff Editor

PLOS ONE

Additional Editor Comments:

Please consider all points raised by reviewers!

Reviewers' comments:

Reviewer's Responses to Questions

**Comments to the Author**

1. If the authors have adequately addressed your comments raised in a previous round of review and you feel that this manuscript is now acceptable for publication, you may indicate that here to bypass the “Comments to the Author” section, enter your conflict of interest statement in the “Confidential to Editor” section, and submit your "Accept" recommendation.

Reviewer #1: (No Response)

Reviewer #3: All comments have been addressed

Reviewer #4: All comments have been addressed

2. Is the manuscript technically sound, and do the data support the conclusions?

Reviewer #1: Partly

Reviewer #3: (No Response)

Reviewer #4: Yes

3. Has the statistical analysis been performed appropriately and rigorously? 

Reviewer #1: I Don't Know

Reviewer #3: (No Response)

Reviewer #4: Yes

4. Have the authors made all data underlying the findings in their manuscript fully available?

Reviewer #1: Yes

Reviewer #3: (No Response)

Reviewer #4: Yes

5. Is the manuscript presented in an intelligible fashion and written in standard English?

Reviewer #1: Yes

Reviewer #3: (No Response)

Reviewer #4: Yes

6. Review Comments to the Author

Reviewer #1: This is a mostly well written paper which analyses retention in oral PrEP through systematic review and modeling.

I have the following suggestions:

- It would be helpful to explain a bit more about the definition of retention, and how this might differ for event driven and daily PrEP. For event driven PrEP- is taking PrEP for each high risk event considered "retained"? So if there were 3 high risk events in 3 months and PrEP was taken each time - is this retained?

- It is not clear why TGP and MSM only were included as there are other groups at risk of HIV not included. This could be better justified in the intro

- Line 107 - 114 in methods - is a little unclear to me. Not sure it follows at line 111 where is says "studies were THEREFORE included....." Maybe delete the therefore since it does not seem to follow. I am also unclear what the definition of retention is as it is not actually stated anywhere - categorizations of retention are described but it is confusing then to include "mixed/other, where composite definitions or a definition other than continuation or follow-up were used."

- Line 116 - not clear what studies conducted in primary care/public health settings means in this context. Suggest sticking to study designs in this section

- Line 117 - it seems odd to me to exclude blinded RCTs. Can you give a reference to support the statement that excluding RCTS means that outcomes approximate real world settings?

- Line 145-149 methods - it would be good to include a variable which allows us to understand whether these studies are from high income countries? I assume most are

- Line 306 - I find "lower loss of PrEP retention rates" hard to understand. Can this be reworded?

- Please include more references in the discussion to support statements. For example in the discussion the authors report that there is poorer retention with more frequent clinic visits - have others also found this? Or is this a novel finding?

- Please discuss more the implication of there being only 1 study of only event driven PrEP and also that 40% of studies did not specify whether it was event driven or daily PrEP.

Reviewer #3: (No Response)

Reviewer #4: Given its strengths, particularly the comprehensive nature of the review and the robustness of the analyses, this manuscript makes a valuable contribution to the understanding of PrEP retention among MSM and TGP. With this revision, it is suitable for acceptance.

7. PLOS authors have the option to publish the peer review history of their article (what does this mean? ). If published, this will include your full peer review and any attached files.

**Do you want your identity to be public for this peer review?** For information about this choice, including consent withdrawal, please see our Privacy Policy .

Reviewer #1: No

Reviewer #3: No

Reviewer #4: No

---

## [Author Response · Author response to Decision Letter 2]

2 Jul 2025

Please refer to the submmitted document with specific comments addressed to the Reviewers.

---

## [Decision Letter · Decision Letter 2]

15 Sep 2025

Oral pre-exposure prophylaxis retention among men who have sex with men and transgender persons: systematic review and meta-analysis

PONE-D-23-22373R2

Dear Dr. Boyd,

Please accept my apologies for the lengthy peer review process, with handling of your manuscript by multiple academic editors. However, I am happy to be able to inform you that your manuscript has been judged scientifically suitable for publication and will be formally accepted for publication once it meets all outstanding technical requirements.

Kind regards,

Steve Zimmerman, PhD

Senior Editor, PLOS One

Additional Editor Comments (optional):

Reviewer #3:

Reviewer #4:

Reviewers' comments:

Reviewer's Responses to Questions

**Comments to the Author**

1. If the authors have adequately addressed your comments raised in a previous round of review and you feel that this manuscript is now acceptable for publication, you may indicate that here to bypass the “Comments to the Author” section, enter your conflict of interest statement in the “Confidential to Editor” section, and submit your "Accept" recommendation.

Reviewer #3: All comments have been addressed

Reviewer #4: All comments have been addressed

2. Is the manuscript technically sound, and do the data support the conclusions?

Reviewer #3: (No Response)

Reviewer #4: Yes

3. Has the statistical analysis been performed appropriately and rigorously? 

Reviewer #3: (No Response)

Reviewer #4: Yes

4. Have the authors made all data underlying the findings in their manuscript fully available?

Reviewer #3: (No Response)

Reviewer #4: Yes

5. Is the manuscript presented in an intelligible fashion and written in standard English?

Reviewer #3: (No Response)

Reviewer #4: Yes

6. Review Comments to the Author

Reviewer #3: (No Response)

Reviewer #4: I thank the Authors for this improved version of their manuscript presenting the points highlighted in the comments. I recommend publishing this revised version of the manuscript.

7. PLOS authors have the option to publish the peer review history of their article (what does this mean? ). If published, this will include your full peer review and any attached files.

**Do you want your identity to be public for this peer review?** For information about this choice, including consent withdrawal, please see our Privacy Policy .

Reviewer #3: No

Reviewer #4: No

---

## [Editor Report · Acceptance letter]

PONE-D-23-22373R2

PLOS ONE

Dear Dr. Boyd,

I'm pleased to inform you that your manuscript has been deemed suitable for publication in PLOS ONE. Congratulations! Your manuscript is now being handed over to our production team.

Kind regards,

on behalf of

Dr Steve Zimmerman

Staff Editor

PLOS ONE